# New criteria for sepsis-induced coagulopathy (SIC) following the revised sepsis definition: a retrospective analysis of a nationwide survey

Toshiaki Iba,[1] Marcello Di Nisio,[2] Jerrold H Levy,[3] Naoya Kitamura,[4] Jecko Thachil[5]

[1]Department of Emergency and Disaster Medicine, Juntendo University Graduate School of Medicine, Tokyo, Japan
[2]Department of Medicine and Ageing Sciences, University G D'Annunzio of Chieti-Pescara, Chieti, Italy
[3]Department of Anesthesiology and Surgery, Duke University School of Medicine, Durham, North Carolina, USA
[4]Department of Recomodulin Strategy Planning, Division of Pharmaceuticals Sales, Asahi Kasei Pharma Corporation, Tokyo, Japan
[5]Manchester Royal Infirmary, Manchester, UK

**Correspondence to**
Dr Toshiaki Iba;
toshiiba@cf6.so-net.ne.jp

## ABSTRACT

**Objective** Recent clinical studies have shown that anticoagulant therapy might be effective only in specific at-risk subgroups of patients with sepsis and coagulation dysfunction. The definition of sepsis was recently modified, and as such, old scoring systems may no longer be appropriate for the diagnosis of sepsis-associated coagulopathy. The aim of this study was to evaluate prognostic factors in patients diagnosed with sepsis and coagulopathy according to the new sepsis definition and assess their accuracy in comparison with existing models.

**Design** Retrospective analysis of the nationwide survey for recombinant human soluble thrombomodulin.

**Setting** General emergency and critical care centres in secondary and tertiary care hospitals.

**Participants** We evaluated the prognostic value of the newly proposed diagnostic criteria for sepsis-induced coagulopathy (SIC). A total of 1498 Japanese patients with sepsis and coagulopathy complications who were treated with recombinant thrombomodulin were analysed in this study.

**Main outcome measures** The platelet count, prothrombin time (PT) ratio, fibrinogen/fibrin degradation products, systemic inflammatory response syndrome score and Sequential Organ Failure Assessment (SOFA) score obtained just before the start of treatment were examined in relation to the 28-day mortality rate.

**Results** The platelet count, PT ratio and total SOFA were independent predictors of a fatal outcome in a logistic regression model. A SIC score was defined using the three above-mentioned variables with a positivity threshold of 4 points or more. The SIC score predicted higher 28-day mortality rate compared with the current Japanese Association for Acute Medicine-disseminated intravascular coagulation score (38.4%vs34.7%).

**Conclusion** The SIC score is based on readily available parameters, is easy to calculate and has a high predictive value for 28-day mortality. Future studies are warranted to evaluate whether the SIC score may guide the decision to initiate anticoagulant therapy.

## Strengths and limitations of this study

► Sepsis-induced coagulopathy (SIC) is the first scoring system specifically designed for coagulation disturbances in sepsis following the new Sepsis-3 definition.
► SIC is defined by the routine coagulation tests such as platelet count and prothrombin time ratio together with Sequential Organ Failure Assessment score.
► Selection bias can exist since all the subjects were treated with recombinant thrombomodulin.
► Evaluation of the prognostic value of the SIC sore in patients receiving no anticoagulant has not yet been performed.

mortality in patients with sepsis independent of the severity of sepsis.[2] Although advances have been made in recent years in the management of septic shock leading to significant improvements in survival, less attention has been paid to the DIC component.[3 4]

Although diagnostic criteria exist for DIC, none were specially designed for sepsis-associated DIC[5–7] that is uniquely characterised by coagulation activation with oversuppression of fibrinolysis and a high incidence of organ dysfunction.[8] The objectives of diagnostic criteria should ideally be to identify a homogenous group of patients with similar pathophysiology and clinical characteristics, whose outcomes can be improved by providing specific treatment interventions.[9] To reach these goals, diagnostic criteria should meet the following three conditions: (1) they should be readily available and easy to use; (2) they should enable diagnostic accuracy and (3) they should have prognostic value. Since the new definition for sepsis was announced in 2016,[10] a DIC score that matches this definition and would help in identifying patients who would benefit from anticoagulant treatment is urgently needed. Therefore, the aim of this study was to evaluate prognostic

## INTRODUCTION

Shock and disseminated intravascular coagulation (DIC) are the two major causes of organ dysfunction in sepsis.[1] Dhainaut *et al* reported that DIC is a strong predictor of

factors in patients diagnosed with sepsis and coagulopathy according to the new sepsis definition and assess their accuracy in comparison with the existing models.

## METHODS
### Data collection
The data set was obtained from a postmarketing survey examining recombinant human-soluble thrombomodulin (TM-α; Asahi Kasei Parma Corporation, Tokyo, Japan) performed by Asahi Kasei Pharma Corporation between May 2008 and March 2010[11] and kindly provided by the Japanese Society on Thrombosis and Hemostasis with permission. All of the cases treated in Japan during this period were registered in the survey. The survey was performed under the supervision of the Japanese Ministry of Health, Labour and Welfare (JMHW) and was conducted in accordance with the Declaration of Helsinki and Good Vigilance Practice and Good Post-marketing Study Practice. Since complete anonymisation of personal data was performed on data collection and the identification of each patient was not possible, the ethical committees waived the need to acquire informed consent from patients.

At the time of data collection, sepsis was defined according to the American College of Chest Physicians/Society of Critical Care Medicine consensus definition.[12] A total of 2516 Japanese patients with sepsis-associated coagulation disorder were registered in this survey; however, since the record of Sequential Organ Failure Assessment (SOFA) data was not mandatory, a complete data set was only obtained in 1498 cases, and all of these patients were analysed in this study. All patients were treated with TM-α (0.06 mg/kg/day for 6 days) by either intravenous bolus injection or intravenous infusion (diluted in 50 mL 0.9% saline) over 15 min via a catheter; the exclusion criteria were as follows: age less than 18 years, major bleeding, systemic inflammatory response syndrome (SIRS) score≦1, hypersensitivity to TM-α and pregnancy.

### Laboratory measurements and organ dysfunction assessments
Blood samples obtained just before the initiation of anticoagulant therapy were analysed, and this data were defined as the 'initial data'. The platelet count, fibrinogen/fibrin degradation products (FDPs) and prothrombin time (PT)-international normalisation ratio were measured in local laboratories. The Japanese Association for Acute Medicine (JAAM)-DIC score[7] was calculated based on the initial (just before TM-α treatment) laboratory data and the SIRS score. Dysfunctions of the respiratory, cardiovascular, hepatic and renal systems were assessed as in the SOFA score.[13] A score of 2 or more within each of these systems was defined as organ dysfunction.

### Statistical analysis
Differences in patient characteristics between survivors and non-survivors were examined using the Fisher's exact test or unpaired Wilcoxon signed-rank test. Then, the relationships between the 28-day mortality rate and the initial data were examined in univariate and multivariate analyses using logistic regression analysis (the enter method). The analysis was conducted using the outcome (survived, 0; died, 1) as the dependent variable and age, sex, SIRS score, platelet count, PT ratio and the proportion of various organ dysfunction as explanatory variables.

The numerical values in the text and tables represent the median and IQR, unless otherwise noted. The results of the logistic regression analysis were reported as the OR, 95% CI and p values. For all the reported results, a value of $p < 0.05$ was considered to denote statistical significance. The above-mentioned analyses were performed using JMP software, V.9.0 (SAS Institute).

## RESULTS
### Baseline characteristics
Among the 1498 patients, 994 (66.4%) survived at 28 days and 504 (33.6%) died. Table 1 shows the baseline characteristics of the patients. The median age was lower and the proportion of women was larger among the survivors. No difference in the SIRS score (p=0.0556) was seen between the two groups, but the JAAM-DIC score (p=0.0019) was higher among the non-survivors. Regarding the haemostatic parameters, the platelet count was higher (p<0.0001) and the PT ratio was lower (p<0.0001) among the survivors. In contrast, the FDP level was not significantly different between the survivors and non-survivors. The initial total SOFA score was significantly higher among the non-survivors (p<0.0001), and the proportions of patients with respiratory dysfunction (p<0.0001), cardiovascular dysfunction (p=0.0426), hepatic dysfunction (p<0.0001) or renal dysfunction (p<0.0001) were higher among the non-survivors.

### Relationships between biomarkers and mortality
The results of the univariate and multivariable analyses are shown in table 2. A multivariate analysis using the enter method showed that patient age (p=0.002), male sex (p=0.017), decreased platelet count (p=0.005), higher PT ratio (p=0.024) and higher total SOFA (p<0.001) were significantly and independently associated with the 28-day mortality rate. In contrast, the initial SIRS score was not correlated with survival (table 2).

### SIC score
The SIC score was developed based on the result of the logistic regression analyses. The relationship between the initial platelet count and mortality is shown in figure 1, A. As shown in the figure, the mortality rate was less than 30% when the platelet count was more than $100 \times 10^9$/L but increased to 35% when the platelet count decreased below $100 \times 10^9$/L. In contrast, the mortality rate increased steadily as the initial PT ratio increased and reached 40% at a PT ratio of more than 1.4 (figure 1, B). The mortality increased along with the increase of the initial total SOFA

**Table 1** Baseline characteristics of the patients

| Characteristics | Survivor (N=994) | Non-survivor (n=504) | p Value |
|---|---|---|---|
| Age (years) | 70 (58–78) | 73 (62–80) | 0.0001 |
| Sex (male/female) | 560/434 | 329/175 | 0.0009 |
| Baseline values | | | |
| SIRS score | 3 (2–4) | 3 (2–4) | 0.0556 |
| SIRS score≧3 | 625 (62.9%) | 341 (67.7%) | 0.0677 |
| JAAM-DIC score | 5 (4–7) | 6 (5–7) | 0.0019 |
| Platelet count (×$10^9$L) | 61 (36–89) | 49 (29–78) | <0.0001 |
| FDP (µg/mL) | 25.3 (13.0–51.9) | 25.4 (12.2–51.7) | 0.7788 |
| PT ratio | 1.30 (1.16–1.48) | 1.36 (1.21–1.64) | <0.0001 |
| Organ dysfunction | | | |
| Total SOFA | 5 (3–6) | 5 (4–7) | <0.0001 |
| Respiratory SOFA≧2* ($PaO_2$/$FiO_2$<300) | 621 (62.5%) | 395 (78.4%) | <0.0001 |
| Cardiovascular SOFA≧2* (requirement of vasopressors) | 636 (64.0%) | 349 (69.3%) | 0.0426 |
| Hepatic SOFA≧2* (bilirubin≧2.0 mg/dL)* | 272 (27.4%) | 202 (40.8%) | <0.0001 |
| Renal SOFA≧2* (creatinine≧2.0 mg/dL) | 326 (32.8%) | 221 (43.8%) | <0.0001 |

Total SOFA is the sum the four items (respiratory SOFA, cardiovascular SOFA, hepatic SOFA, renal SOFA). The score of total SOFA is defined as 2 if the total score exceeded 2.
*Defined according to the Third International Consensus for Sepsis and Septic Shock.[13]
DIC, disseminated intravascular coagulation; FDP, fibrinogen/fibrin degradation product; $FiO_2$, fractional inspired oxygen; JAAM, Japanese Association for Acute Medicine; $PaO_2$, partial pressure of oxygen; PT, prothrombin time; SIRS, systemic inflammatory response syndrome; SOFA, Sequential Organ Failure Assessment.

(figure 1, C). For the SIC scoring, the organ dysfunction scores were defined according to the criteria used for the SOFA score.[13] Total SOFA is calculated as the sum of the four items (respiratory SOFA, cardiovascular SOFA, hepatic SOFA, renal SOFA). The score of total SOFA was defined as 2 if the total score exceeded 2. For the PT ratio, the cut-off value to assign 1 point was set at 1.2 in accordance with the JAAM-DIC criteria, while the cut-off value for 2 points was set at 1.4 based on the case number distribution and the mortality rate. Finally, SIC was defined as a total score of 4 or more (table 3), since the mortality rate at this point exceeded 20% (figure 2, A), with the requirement that the total score for the platelet count and the PT ratio exceed 2.

### Comparison of the SIC and JAAM-DIC criteria

A total of 902 patients were diagnosed as having SIC, while 1332 patients met the JAAM criteria for DIC. The respective mortality rates for these classifications were 38.4% and 34.7%. Among the patients diagnosed with DIC by the JAAM score, 477 cases were negative with SIC, while 47 patients were DIC negative using the JAAM-DIC but positive with SIC (table 4). The mortality of the patients having positive JAAM-DIC but negative SIC was 27.7%, while that of patients positive with SIC but negative with JAAM-DIC was 34.0%. Figure 2 shows the relationship between the 28-day mortality rate and the SIC score (left panel) and the JAAM-DIC score (right panel). Using the SIC scoring system, the mortality rate increased

**Table 2** Relationship between 28-day mortality and baseline characteristics

| Item | Univariate | | | Multivariate | | |
|---|---|---|---|---|---|---|
| | OR | 95% CI | p Value | OR | 95% CI | p Value |
| Age (years) | 1.010 | 1.004 to 1.016 | 0.002 | 1.010 | 1.004 to 1.017 | 0.002 |
| Sex (male/female) | 1.457 | 1.168 to 1.821 | 0.001 | 1.323 | 1.052 to 1.668 | 0.017 |
| Platelet count (×$10^9$L) | 0.965 | 0.945 to 0.984 | 0.000 | 0.972 | 0.951 to 0.992 | 0.005 |
| PT ratio | 1.225 | 1.065 to 1.435 | 0.004 | 1.169 | 1.020 to 1.364 | 0.024 |
| Total SOFA | 1.252 | 1.181 to 1.328 | 0.000 | 1.213 | 1.143 to 1.289 | <0.001 |

Total SOFA is the sum the four items (respiratory SOFA, cardiovascular SOFA, hepatic SOFA, renal SOFA). The score of total SOFA is defined as 2 if the total score exceeded 2.
PT, prothrombin time; SOFA, Sequential Organ Failure Assessment.

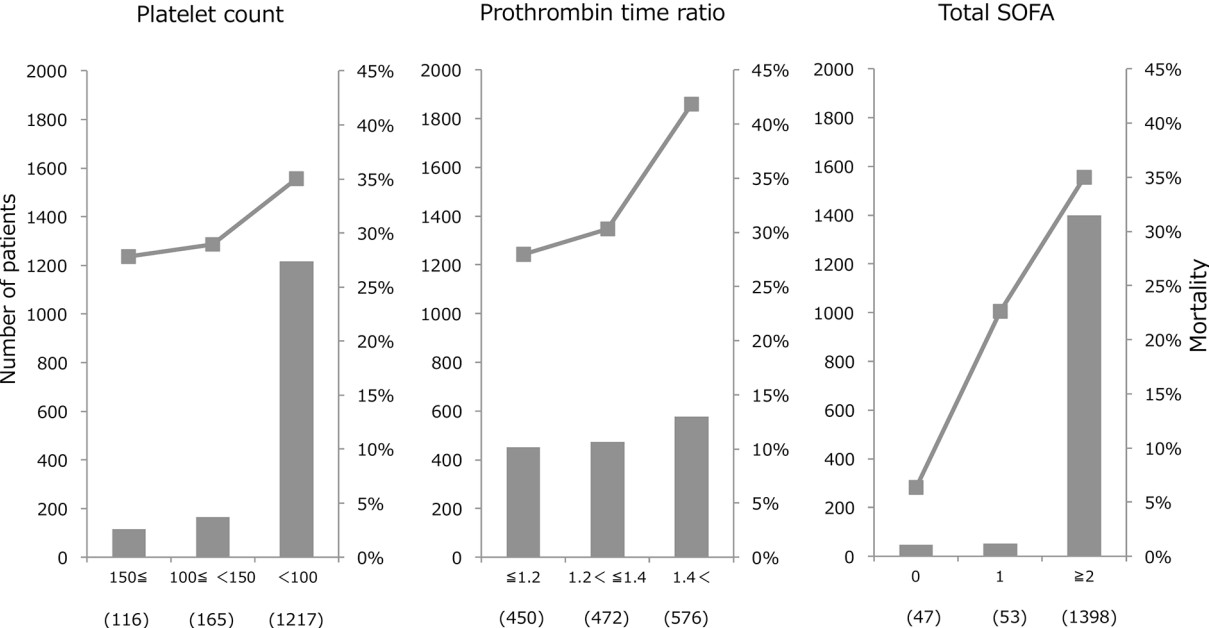

**Figure 1** Patient counts and mortality rates according to platelet count, prothrombin time ratio and total Sequential Organ Failure Assessment (SOFA) score. The bar graph shows the number of patients in each category, and the line graph represents the mortality rate. The x-axis represents the score and (case number). A: many of the patients had an initial platelet count of $100 \times 10^9$/L or less. Mortality increased to 35% when the count decreased to less than $100 \times 10^9$/L. B: mortality increased along with an increase in the prothrombin time ratio, reaching more than 40% when the prothrombin time ratio was more than 1.4. C: the population of total SOFA score of 0 and 1 is quite limited, and the mortality of this population was lower than that of the score of 2 or more.

in a linear fashion. The mortality rate was 30% at a score of 4 and increased steeply to a maximum of over 45% at a score of 6. In contrast, the mortality rate did not seem to be strongly correlated with the JAAM-DIC score. The mortality rate had already exceeded 20% at scores of 1–3, and it increased to over 30% at a score of 4. The mortality rate increased only gradually thereafter, reaching approximately 40% at a score of 8.

The median JAAM-DIC score in the survivors was 5 (4–7) before treatment and it decreased to 3 (1–4). The score also decreased from 6 (5–7) to 5 (4–6) in the non-survivors. In contrast, though SIC score decreased

from 5 (4–5) to 3 (2–3) in survivors, it did not decrease in non-survivors (5 (4–6) to 5 (3–5)).

## DISCUSSION

The relationship and balance between coagulation and fibrinolysis system is regulated by a complex series of interactions. Microbial products induce the synthesis and the release of inflammatory mediators called pathogen-associated molecular patterns (PAMPs) during sepsis. In addition to these PAMPs, proinflammatory substances known as damage-associated molecular patterns (DAMPs) are also released from activated or damaged host cells.[14 15] Both PAMPs and DAMPs activate the coagulation system together with the host immune response, leading to DIC.[14 16] Since the primary function of these biologic responses is to assist the host in sequestering and eliminating the microbes, the suppression of coagulation at this stage may not always lead to a better outcome. In contrast, patients with compromised organ circulation may benefit from anticoagulant therapy.[17] However, all of the anticoagulants examined in the early 21st century failed to show any efficacy in these patients.[18 19] One of the major reasons for this was that these studies targeted severe sepsis (without coagulation disturbances), rather than sepsis-associated coagulopathy.[2 20] After a series of disappointing reports, Umemura *et al* suggested that anticoagulant therapies might be beneficial if they were applied to patients with sepsis who have a severe coagulation disorder.[21] In addition, Yamakawa *et al* also reported

**Table 3** Scoring for the diagnosis of sepsis-induced coagulopathy

| Category | Parameter | 0 point | 1 point | 2 points |
|---|---|---|---|---|
| Prothrombin time | PT-INR | ≦1.2 | >1.2 | >1.4 |
| Coagulation | Platelet count (×10⁹/L) | ≧150 | <150 | <100 |
| Total SOFA | SOFA four items | 0 | 1 | ≧2 |

Diagnosed as sepsis-induced coagulopathy when the total score is 4 or more with total score of prothrombin time and coagulation exceeding 2.
Total SOFA is the sum the four items (respiratory SOFA, cardiovascular SOFA, hepatic SOFA, renal SOFA). The score of total SOFA is defined as 2 if the total score exceeded 2.
INR, international normalisation ratio; PT, prothrombin time; SOFA, Sequential Organ Failure Assessment.

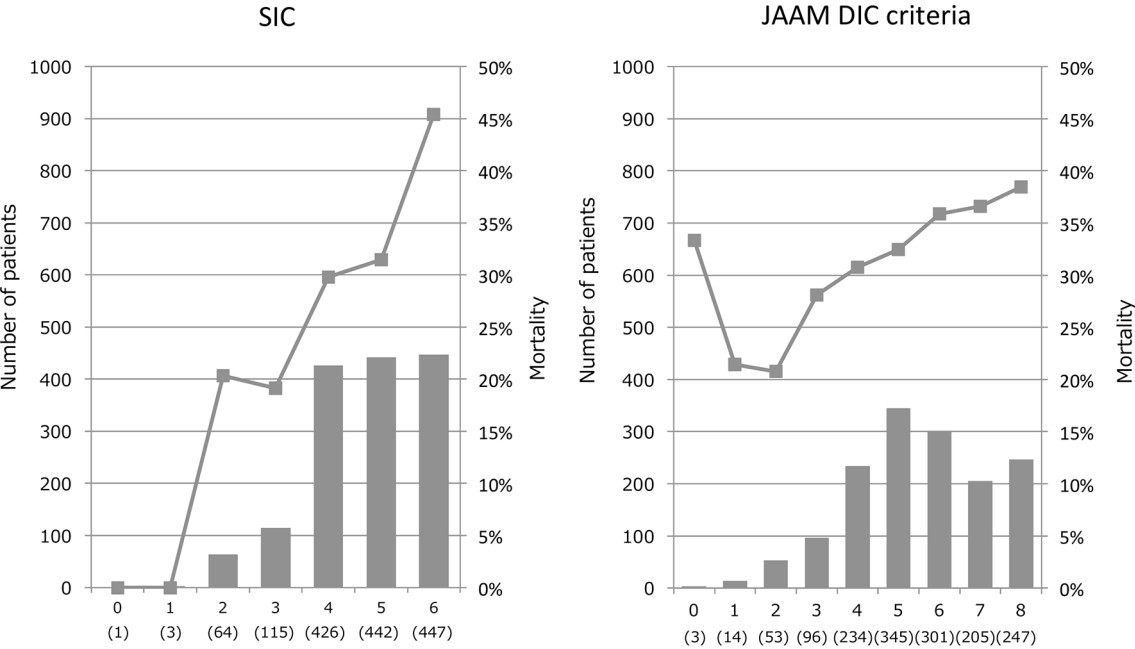

**Figure 2** Patient counts and mortality rates according to the SIC and the JAAM-DIC classifications. The patient distributions (bars) and the mortality rates (lines) are plotted according to the SIC scores (left) and the JAAM-DIC scores (right). The x-axis represents the score and (case number). The mortality rate increased as SIC score elevated and exceeded 20% at a score of 4. In contrast, the mortality rate exceeded 30% at a JAAM-DIC score of 4 and gradually increased to 40%. DIC, disseminated intravascular coagulation; JAAM, Japanese Association for Acute Medicine; SIC, sepsis-induced coagulopathy.

an association between the application of anticoagulant therapy and a decrease in mortality among patients with a high risk of death (SOFA score of 13–17).[22] With respect to recombinant thrombomodulin, recent studies examining its effects in patients with septic DIC have repeatedly reported favourable results[2 22] with more prominent benefits as the severity of coagulopathy increased.[23 24] In a phase II clinical trial performed in 17 countries, recombinant thrombomodulin tended to exhibit an effect in patients with organ dysfunction having a PT ratio of greater than 1.4 before the treatment.[24] Recombinant thrombomodulin has been used in Japanese patients who meet the JAAM-DIC criteria, which, however, might not be appropriate.[25] Yoshimura et al[26] reported that a beneficial effect was only seen in patients diagnosed as having DIC and had an APACHE II score of between 24 and 29. Thus, to identify appropriate candidates for anticoagulant therapy, we added a category for organ dysfunction in patients with SIC and targeted individuals with a mortality rate higher than that of patients meeting the JAAM-DIC

criteria. This new SIC category represents 'sepsis (Third International Consensus Definitions (Sepsis 3)[10]) and coagulation disorder,' which could be a suitable target for therapeutic interventions. Previous reports have consistently reported efficacy when the 30-day mortality rate in the TM-α-treated group was between 20% and 30%,[27 28] and not below 20%.[24] As a matter of fact, a subgroup analysis of a phase III study performed in Japan demonstrated that the mortality rate in a heparin-control group was 31.6%, while that of TM-α group was 21.4%.[29] In the present study, the mortality rate of the SIC group was about 30% when the total score was 4, and it increased as the SIC score increased.

Another purpose of this study was to compare the SIC and JAAM-DIC criteria. In SIC, the SIRS score used in the JAAM-DIC criteria was replaced with the SOFA score, and the FDP criterion was eliminated. This was based on the fact that the prognostic relevance of SIRS has been questioned and was not used in the new definition of sepsis. Besides, the significance of fibrin-related markers in DIC differs depending on the underlying disease,[30] and the impact of the FDP criterion was limited to patients with fibrinolysis-suppressed-type coagulation disorders, as represented by sepsis. The current study indicated that the SIC score identified patients with a higher risk of death, compared with the JAAM-DIC score (38.4% vs 34.7%), and the numbers of diagnosed cases were 902 and 1332, respectively. The mortality of patients who satisfied SIC criteria but not JAAM-DIC criteria was 6.3% higher than those diagnosed using JAAM-DIC criteria that did not satisfied SIC criteria. Therefore, we speculate that

**Table 4** Patient count and mortality

|  |  | SIC | | |
|---|---|---|---|---|
|  |  | + | − | Total |
| JAAM-DIC | + | 855 (38.6%) | 477 (27.7%) | 1332 (34.7%) |
|  | − | 47 (34.0%) | 119 (21.8%) | 166 (25.3%) |
|  | Total | 902 (38.4%) | 596 (26.5%) | 1498 (33.6%) |

JAAM-DIC, Japanese Association for Acute Medicine-disseminated intravascular coagulation; SIC, sepsis-induced coagulopathy,

the SIC score might better identify candidates for anticoagulant therapy rather than the JAAM-DIC score. The other advantage of SIC is the simplicity of its calculation. Indeed, the FDP criterion was omitted, and SIRS score was replaced by the SOFA score. Since the SOFA score is routinely evaluated in the ICU, the addition of a PT test is relatively easy to implement in clinical practice.

## Limitations

There are limitations to our current study. First, data from a postmarketing survey were used in this study and all subjects were treated with recombinant thrombomodulin. While treatment could influence the overall 28-day mortality rate, it is unlikely that it affected the performance of the SIC score. This study also did not analyse the ability of SIC in identifying patients with sepsis with coagulopathy not treated with any anticoagulants or treated with anticoagulants other than TM-α, such as antithrombin.

## CONCLUSIONS

We have proposed SIC as a new score for patients with 'sepsis and coagulopathy.' Since the SIC score adheres to the new sepsis criteria (Third International Consensus Definitions), this definition will be easy to use and should provide important and novel information to the physicians.

**Acknowledgements** The authors thank all the institutes that participated in the postmarketing surveillance.

**Contributors** TI, MDN and JT conceived the study and participated in its design. TI and MDN participated in the sequence alignment and drafted the manuscript. JL helped to revise the manuscript. NK helped to collect and arrange the data. All authors read and approved the final manuscript.

**Funding** This work was supported by Ministry of Education, Culture, Sports, Science and Technology-Supported Program for the Strategic Research Foundation at Private Universities 2016.

**Competing interests** NK is an employee of Asahi Kasei Pharma.

**Provenance and peer review** Not commissioned; externally peer reviewed.

**Data sharing statement** Extra data can be accessed via the Dryad data repository at http://datadryad.org/ with the doi:10.5061/dryad.3ds63.

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
