## [Reviewer comments · BMJ Open]

ARTICLE DETAILS

TITLE (PROVISIONAL)	New Criteria for Sepsis-induced coagulopathy (SIC) following the revised sepsis definition: a retrospective analysis of a nationwide survey.
AUTHORS	Iba, Toshiaki ; Di Nisio, Marcello; Levy, Jerrold; Kitamura, Naoya; Thachil, Jecko

VERSION 1 - REVIEW

REVIEWER	Barbara Adamik Medical University Dept. of Anaesthesiology and Intensive Therapy, Poland
REVIEW RETURNED	09-Jun-2017

GENERAL COMMENTS	Thank you for the opportunity to review this important study. It is a well and clearly written evaluation of prognostic factors in patients diagnosed with sepsis and coagulopathy. The analysis included 1498 patients. Authors proposed a sepsis induced coagulation (SIC) score which might be helpful to identify septic patients for whom anticoagulant therapy would be the most beneficial. The authors may wish to consider the following comments: 1, Results section - Comparison of the SIC and JAAM-DIC criteria. Please describe in details observed differences in JAAM DIC and SIC presented in Table 4: 477 patients were evaluated as positive with JAAM DIC but negative with SIC; 47 patients were evaluated as negative with JAAM DIC but positive with SIC. All these differences should be also discussed in the Discussion section.2. Table 1: please clarify "The score of each organ is defined as 2 in case of 2 or more"3. According to the section Laboratory measurements and organ dysfunction assessments, blood samples for analysis were obtained just before the initiation of anticoagulant therapy. So the limitation that all subjects were treated with recombinant thrombomodulin applies to the 28-day mortality rate but not to the potential use of SIC for the identification of septic patients with a risk of coagulopathy.
--

REVIEWER	Cheng-Ming Tsao Taipei Veterans General Hospital, Taiwan
REVIEW RETURNED	14-Jun-2017

GENERAL COMMENTS	The data describe new and interesting criteria for sepsis-induced coagulopathy (SIC). The manuscript is clear and well written. There are some minor comments:  1. All data utilized in this study was from a post-marketing survey and all the subjects were treated with recombinant thrombomodulin. Thus, it is better to show the changes in your SIC scores through 28 days compared to the JAAM-DIC criteria for evaluating the efficacy of anticoagulant treatment. 2. Total SOFA in your SIC scores is scored based on the 4 scores of respiratory, cardiovascular, hepatic and renal SOFA. The score of each organ is defined as 2 in case of 2 (shown in Table 1 legend). Thus, why the score of total SOFA is defined as 2 if the total score exceeded 2, not higher? 3. The SOFA scores generally include neurological coma score. Why this neurological SOFA score is not involved in your SIC scores?
---

VERSION 1 – AUTHOR RESPONSE

Reviewer: 1

Reviewer Name: Barbara Adamik

Institution and Country: Medical University Dept. of Anaesthesiology and Intensive Therapy, Poland

Please state any competing interests or state 'None declared': None declared

The authors may wish to consider the following comments:

1. 1, Results section - Comparison of the SIC and JAAM-DIC criteria.

Please describe in details observed differences in JAAM DIC and SIC presented in Table 4: 477 patients were evaluated as positive with JAAM DIC but negative with SIC; 47 patients were evaluated as negative with JAAM DIC but positive with SIC. All these differences should be also discussed in the Discussion section.

Reply

Thank you for the valuable comments. We added "Among the patients diagnosed with DIC by the JAAM score, 477 cases were negative with SIC while 47 patients were DIC negative using the JAAM-DIC but positive with SIC" (Page 10, Line 6-8), and "The mortality of the patients having positive JAAM-DIC but negative SIC was 27.7%; while that of patients positive with SIC but negative with JAAM-DIC was 34.0%." (Page 10, Line 8-10) In addition, we added "The mortality of the patients having positive with SIC but negative with JAAM-DIC was 6.3% higher than that of the patients positive with JAAM-DIC but negative with SIC." in Discussion section. (Page 13, Line 10-11)

2. Table 1: please clarify “The score of each organ is defined as 2 in case of 2 or more”

Reply

Thank you for the suggestion. We have added “Total SOFA is calculated as the sum of the 4 items (respiratory SOFA, cardiovascular SOFA, hepatic SOFA, renal SOFA). The score of total SOFA was defined as 2 if the total score exceeded 2.” in results section (Page 9, Line 15-16) and “The score of total SOFA is defined as 2 if the total score exceeded 2” in the legends of Table 1, 2 and 3. We also have clarified the international criteria used to define to SOFA items as reported in the Table and added the reference in the legend.

3. According to the section Laboratory measurements and organ dysfunction assessments, blood samples for analysis were obtained just before the initiation of anticoagulant therapy. So the limitation that all subjects were treated with recombinant thrombomodulin applies to the 28-day mortality rate but not to the potential use of SIC for the identification of septic patients with a risk of coagulopathy.

Reply

We thank the reviewer for the suggestion and have clarified this point in the discussion. We have re-phrased this section as follows. “While treatment could influence the overall 28-day mortality rate, it is unlikely that it affected the performance of the SIC score. The potential use of SIC for the identification of septic patients with coagulopathy should also be examined in patients not treated with anticoagulants, or treated with other anticoagulants, such as antithrombin.” (Page 13, Line 20-Page 14, Line 3)

Reviewer: 2

Reviewer Name: Cheng-Ming Tsao

Institution and Country: Taipei Veterans General Hospital, Taiwan

Competing Interests: None declared.

There are some minor comments:

1. All data utilized in this study was from a post-marketing survey and all the subjects were treated with recombinant thrombomodulin. Thus, it is better to show the changes in your SIC scores through 28 days compared to the JAAM-DIC criteria for evaluating the efficacy of anticoagulant treatment.

Reply

Thank you for the suggestion. We added the following sentences in the results section: “The median JAAM-DIC score in the survivors was 5 (4 to 7) before treatment and it decreased to 3 (1 to 4). The

score was also decreased from 6 (5 to 7) to 5 (4 to 6) in the non-survivors. In contrast, though SIC score decreased from 5 (4 to 5) to 3 (2 to 3) in survivors, it did not decrease in non-survivors (5 [4 to 6] to 5 [3 to 5])." (Page 10, Line 17-20)

2. Total SOFA in your SIC scores is scored based on the 4 scores of respiratory, cardiovascular, hepatic and renal SOFA. The score of each organ is defined as 2 in case of 2 (shown in Table 1 legend). Thus, why the score of total SOFA is defined as 2 if the total score exceeded 2, not higher?

Reply

Thank you for the valuable comments. SOFA score represents the organ dysfunction and not the coagulation disorder. Since we defined SIC as sepsis (infection-induced organ dysfunction) with coagulation disorder, the purpose of the inclusion of SOFA is just the confirmation of organ dysfunction. Thus, the maximum score was limited to 2 similar to the other coagulation factors.

3. The SOFA scores generally include neurological coma score. Why this neurological SOFA score is not involved in your SIC scores?

Reply

Thank you for the valuable comments. We are aware that the SOFA score includes neurological coma score. However, we did not include the nervous system SOFA for some reasons. First, the neurological SOFA is often difficult to assess accurately and it would add complexity to the calculation of SIC. Secondary, the relevance of this item to the coagulation dysfunction is unclear. While a microcirculatory disturbance caused by thrombus formation may play major roles in the development of renal, cardiovascular, hepatic and respiratory dysfunctions, a direct involvement in the neurological dysfunction is not clear.

VERSION 2 – REVIEW

REVIEWER	Barbara Adamik Medical University, Department of Anaesthesiology and Intensive Therapy, Wroclaw, Poland
REVIEW RETURNED	13-Jul-2017
GENERAL COMMENTS	In my opinion the manuscript is suitable for publication in BMJ Open. Congratulations to the Authors.

REVIEWER	Cheng-Ming Tsao Department of Anesthesiology, Taipei Veterans General Hospital, Taiwan
REVIEW RETURNED	07-Jul-2017

GENERAL COMMENTS	Well Done.
------------